# Revisiting Unsupervised Temporal Action Localization: The Primacy of High-Quality Actionness and Pseudolabels

## ABSTRACT

Recently, temporal action localization (TAL) methods, especially the weakly-supervised and unsupervised ones, have become a hot research topic. Existing unsupervised methods follow an iterative "clustering and training" strategy with diverse model designs during training stage, while they often overlook maintaining consistency between these stages, which is crucial: more accurate clustering results can reduce the noises of pseudolabels and thus enhance model training, while more robust training can in turn enrich clustering feature representation. We identify two critical challenges in unsupervised scenarios: **1. What features should the model generate for clustering? 2. Which pseudolabeled instances from clustering should be chosen for model training?** After extensive explorations, we proposed a novel yet simple framework called Consistency-Oriented Progressive high actionness Learning to address these issues. For feature generation, our framework adopts a High Actionness snippet Selection (HAS) module to generate more discriminative global video features for clustering from the enhanced actionness features obtained from a designed Inner-Outer Consistency Network (IOCNet). For pseudolabel selection, we introduces a Progressive Learning With Representative Instances (PLRI) strategy to identify the most reliable and informative instances within each cluster for model training. These three modules, HAS, IOCNet, and PLRI, synergistically improve consistency in model training and clustering performance. Extensive experiments on THUMOS'14 and ActivityNet v1.2 datasets under both unsupervised and weakly-supervised settings demonstrate that our framework achieves the state-of-the-art results.

## CCS CONCEPTS

• **Information systems** → *Video search*.

## KEYWORDS

Multimodal Understanding, Unsupervised Temporal Action Localization, Progressive Learning, Consistency Constraint.

## 1 INTRODUCTION

The proliferation of intelligent surveillance devices in recent years has necessitated the development of efficient video processing techniques in the multimedia understanding field [19, 40, 41]. Among

*ACM MM, 2024, Melbourne, Australia*
© 2024 Copyright held by the owner/author(s). Publication rights licensed to ACM.
ACM ISBN 978-x-xxxx-xxxx-x/YY/MM
https://doi.org/10.1145/nnnnnnn.nnnnnnn

them, temporal action localization (TAL) [16, 38], which aims to accurately localize the temporal boundaries of actions in untrimmed videos and identify their categories from a pre-defined action list, is one of the most important areas.

In initial research stages, temporal action localization methods predominantly follow a fully supervised setting [5, 27, 56], which requires the annotator to perform segment-level annotations for each video in the training dataset, i.e., annotate all action instances with the start and end timestamps and the corresponding action categories. This process requires a lot of manual operations and is prone to the problems of costly annotation and subjectivity of the annotation results. To address those issues, weakly supervised temporal action localization (WTAL) [14, 39, 53] has gradually emerged as a new research focus, which only requires the video-level annotation, i.e., labeling all the pre-defined action categories contained in each video. However, compared to simply collecting unlabeled videos, even such video-level annotations require a significant cost.

As a result, the focus has shifted towards training video action localization networks on unlabeled videos, a task known as Unsupervised Temporal Action Localization (UTAL) [10]. This method only requires annotating the total number of action categories for the entire dataset, thereby further reducing annotation costs. As illustrated in Fig 1(a), all existing UTAL methods follow an iterative "clustering-training" pipeline [10, 48]. During each iteration, the video-level pseudolabels for each video are generated based on the clustering results of the global video features, which are derived from class-agnostic attention obtained from a localization model $\mathcal{M}$. Subsequently, the entire pseudolabeled video set is adopted to train the localization model $\mathcal{M}$.

While the above unsupervised methods have garnered notable success, their primary emphasis lies in the "training" stage, i.e., designing various localization models based on uncertainty [48] or co-attention mechanisms [10, 26], while ignoring the high consistency between the "clustering" and "training" stages. This consistency is crucial for improving clustering accuracy and enhancing model training robustness during iterations, because accurate clustering reduces pseudolabel noise while robust training enriches clustering feature representation. This raises two key challenges: **1) What features should the model generate for clustering? 2) Which pseudolabeled instances from clustering should be chosen for model training?** After extensive investigations, we argue that global video features with high actionness across two-branch and high-quality pseudolabels matter for these two challenges. For feature generation, since existing methods aggregate global video features using only the class-agnostic attention of the entire video, the importance of high actionness snippets and the inconsistency of action positioning between class-agnostic and class-specific branches are often ignored. For pseudolabel selection, adopting all pseudolabeled videos for training is not advisable since the clustering results are unreliable in the initial iteration.

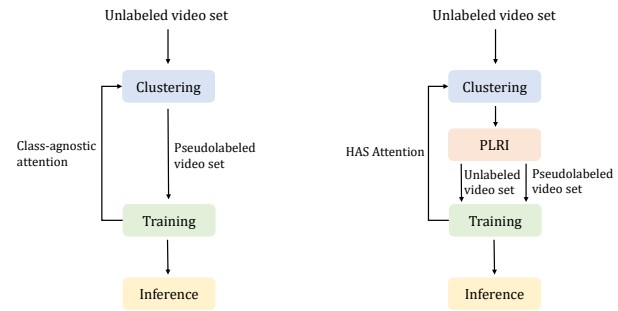

(a) The pipeline of existing methods     (b) The pipeline of our COPL

**Figure 1: Comparison between the pipelines of existing methods and COPL. Compared to existing methods, our COPL employs a progressive learning strategy to divide the video collection into two subsets, simultaneously utilizing HAS attention for iterative refinement.**

To address the above issues, we propose a novel yet simple **C**onsistency-**O**riented **P**rogressive **L**earning (COPL) framework for UTAL. Specifically, the **H**igh **A**ctionness snippet **S**election (HAS) clustering module utilizes a two-branch consistency criterion to generate filtered HAS attention, which eliminates numerous background segments while retaining only snippets with discriminative action-related information for global video feature generation. Subsequently, to generate the high-quality pseudolabeled video sets, we introduce a progressive learning with representative instances strategy based on intra-cluster cohesion and inter-cluster separation, aiming to identify and exclude video instances exhibiting comparatively lower discriminative video global features. During the training phase, utilizing both the refined pseudolabeled video set and the unlabeled video set, we design an **I**nner-**O**uter **C**onsistency **Net**work (IOCNet) based on the Teacher-Student architecture [42]. This network enforces consistency constraints jointly from modalities, branches, and data augmentation, thereby enhancing discriminative learning capabilities without relying on explicit labels. The entire framework's workflow is illustrated in Fig 1(b). As depicted in the figure, the three crucial modules progress sequentially through iterative optimization, reinforcing the discriminative features and effectively improving model performance.

The contributions of this paper are as follows:

- We propose a conceptually novel yet simple framework termed Consistency-Oriented Progressive Learning, to tackle two crucial challenges, i.e., feature generation and pseudolabel selection, of unsupervised temporal action localization.
- Our framework presents three crucial modules: a high actionness snippet selection module that generates high actionness global video feature, a PLRI strategy that excludes low-confidence video instances based on clustering distributions, and an IOCNet model with enriched actionness-aware abilities via multiple consistency constraints.
- We extensively evaluate our COPL framework on the THUMOS'14 and ActivityNet v1.2 datasets. The results demonstrate that COPL achieves state-of-the-art performance.

## 2 RELATED WORK

### 2.1 Weakly-supervised Temporal Action Localization

Weakly supervised Temporal Action Localization (WTAL) has raised great research attention recently, owing to its relatively lower annotation costs. Existing methods can be categorized into two groups: representation learning based methods and temporal attention modeling methods. The first group designs various loss functions for better feature representations [8, 10], such as the multi-label center loss [30] that penalizes the distance between the features and the corresponding action class centers and the snippet-level contrastive loss [12, 53] that refines the representations of the action snippets. The second group aims to enrich the class-agnostic attention weights to separate the action and the background parts [15, 29]. For example, Lee et al. [20] added a suppression branch to the network, effectively modeling background class as an auxiliary component for WTAL. Besides, Li et al. [24] identified the actionness inconsistency in a classical two-branch network and then introduce an action consistency loss to mitigate this issue.

As previously discussed, the UTAL method iteratively employs pseudolabels obtained through clustering for training WTAL models. However, current WTAL models face two challenges if adopted in unsupervised settings: handling noisy instances and effectively utilizing both pseudolabeled and unlabeled video datasets. In contrast, our IOCNet tackles these challenges by introducing a Teacher-Student network and employing several inner-outer consistency losses, thereby enhancing the model's ability to mine high actionness snippets and resist noisy instances.

### 2.2 Unsupervised Temporal Action Localization

Recent advances have shifted into unsupervised temporal action localization task. The prevailing UTAL approaches adopt an iterative pipeline, i.e., first adopting spectral clustering [37] to generate action pseudolabel for each video, and then training a localization model with pseudolabels, which is adopted to generate the pseudolabels for next iteration. The principal difference among these methods lies in the second phase, namely the design of the localization model. Some methods leveraged representation learning techniques to distinguish between action and background in the video [26] or manipulate the distribution within the embedding space [10]. In contrast, other approaches incorporated uncertainty awareness to enhance the learning process for both RGB and optical flow features [48].

In conclusion, these methods uniformly generate the global feature only with the class-agnostic branch and utilize all pseudolabeled instances for training. These operations, however, do not filter out noisy instances during clustering, nor do they enrich the high actionness expression of video features. In contrast, our COPL effectively addresses these issues with three coupled modules.

### 2.3 Progressive Learning

Progressive learning (PL) aims to train models incrementally, guiding them from simpler instances to more complex ones. This approach has found widespread use in representation learning [6, 55], image classification [7, 57], and ReID [9, 13, 47]. There are two

primary types of PL methods. The first one concentrates on the network level [11, 57]. These methods aid in mitigating the challenges of training deep networks by progressively enhancing network capacity (both width and depth) [21, 54]. This enhancement is crucial for improving the abilities of the network. The second type focuses on the data level [43, 47]. These methods leverage PL to gradually integrate a substantial amount of unlabeled data into the training process within an incomplete-supervised setting, thus addressing the scalability challenge of incomplete labeled datasets [2, 9].

In this paper, we draw inspiration from these methods and incorporate the concept of progressive learning into UTAL. The distinctions are as follows: 1) we design a novel confidence measurement criterion based on the fully connected graph employed in spectral clustering, which is effective in assessing the reliability of the video instances; 2) we train pseudolabeled video set and unlabeled video set with IOCNet jointly. This approach maximally utilizes information from all instances.

## 3 METHODOLOGY

### 3.1 Preliminaries

We first detail the denotations in this section. Under the UTAL settings, given the unlabeled videos set $V = \{X_1, X_2, \ldots, X_N\}$ that contains $N$ videos from $C$ distinct action categories, the goal is to train a localization model $\phi(\theta; \cdot)$ parameterized with $\theta$ to directly classify and localize the corresponding action segments in an untrimmed video. For a video $X$ of length $T$ in $V$, we first adopt the pre-trained network to extract its RGB features $X^R = \{x_i^R\}_{i=1}^T \in R^{T*d}$ and optical flow features $X^F = \{x_i^F\}_{i=1}^T \in R^{T*d}$, which are also concatenated together to obtain $X = \{x_i\}_{i=1}^T \in R^{T*2d}$. As illustrated in Fig 2, taking the above features of each video in $V$ as input, our COPL contains the following stages during each iteration: 1) The HAS module generates the high-actionness attention $A^{HAS}$ that exhibit consistency across both the class-specific branch $A_{cs}$ and class-agnostic branch $A_{ca}$ of IOCNet to obtain the discriminative global feature $F$ for each video; 2) The PLRI module performs clustering on all video features, and generates the pseudolabeled set $P$ and unlabeled video set $U$, based on the intra-cluster cohesion and inter-cluster separation; 3) The IOCNet is trained with the $P$ and $U$ sets, which facilitates the generation of $A^{HAS}$ in next iteration. We will introduce these three stages in detail.

### 3.2 High Actionness Snippet Selection

During the clustering stage, existing UTAL methods typically employ the snippet-wise attention of an entire untrimmed video to generate its global video feature. This attention is often derived from a single branch, such as the class-agnostic branch or the class-specific branch aggregated along the action category dimension. However, there are two serious problems with obtaining high actionness features for clustering: 1) Using complete attention is not advisable. Untrimmed videos typically contain substantial background snippets. Including these snippets in the attention may decrease the action-specific expression of the global feature. 2) Results obtained from a single branch may lack reliability. As shown in Fig 3, we visualize the activation zones of high and low activation in two branches. It can be observed that class-specific branch

is prone to interference from contextual information, leading to erroneously high activation zones when significant background related to the action exists. Conversely, the class-agnostic branch is susceptible to interference from actions unrelated to the target, resulting in erroneously high activation zones. Both scenarios can lead to inconsistent and unexpected localization results [24, 28].

To this end, we propose the HAS module based on the principle of two-branch consistency, which enriches the reliable actionness of global feature extracted from a video. This module selects high-actionness snippets that exhibit consistency across both branches to overcome the limitations of relying on the complete attention of a single branch. With the filtered discriminative snippets, the quality of global video feature and the accuracy of later clustering can be enhanced.

Specifically, as shown in Fig 2(b), the pre-trained RGB and Flow features of a video are fed into the Teacher network of IOCNet to obtain the attention $A_{cs} = \{a_{cs,i}\}_{i=1}^T$ and $A_{ca} = \{a_{ca,i}\}_{i=1}^T$ from the class-specific branch and class-agnostic branch, respectively. These two attention scores are combined together as actionness-aware attention $A_c = \{a_{c,i}\}_{i=1}^T$, $a_{c,i} = \frac{a_{ca,i}+a_{cs,i}}{2}$. Thereafter, an action filter is performed to obtain high-actionness attention across two branch. Firstly, a binarization operation on $A_{cs}$ and $A_{ca}$ is conducted to generate high activation zones and low activation zones [52] on two branches as:

$$a_i^{bin} = \begin{cases} 1, & \text{if } a_i > \text{median}(A) \\ 0, & \text{otherwise} \end{cases} \quad (1)$$

where $median(A)$ is the median value of actionness sequence $A$. To identify and filter out the snippets with low actionness across two branches, a consistency comparison between $A_{cs}^{bin}$ and $A_{ca}^{bin}$ is conducted as:

$$a_{c,i}^f = \begin{cases} a_{c,i}, & \text{if } a_{ca,i}^{bin} = a_{cs,i}^{bin} = 1 \\ 0, & \text{otherwise} \end{cases} \quad (2)$$

Thereafter, we obtain the actionness-aware attention $A_c^f = \{a_{c,i}^f\}_{i=1}^T$. Subsequently, the top-$k$ snippets of $A_c^f = \{a_{c,i}^f\}_{i=1}^T$ is selected to generate the high-actionness attention as:

$$a_i^{HAS} = \begin{cases} a_{c,i}^f, & \text{if } i \in A_c^{f,Desc}[:k] \\ 0, & \text{otherwise} \end{cases} \quad (3)$$

where $A_c^{f,Desc}$ represents the index of $A_c^f$ when sorted in descending order. $k = max(1, \lfloor \frac{T}{\gamma} \rfloor)$. $\gamma$ controls the proportion of selected snippets. The obtained attention $A^{HAS} = \{a_i^{HAS}\}_{i=1}^T$ combines key snippets of the video from both the class-specific and class-agnostic branches, emphasizing action-discriminative snippets. However, this action filter may be overly aggressive, leading to the removal of crucial action snippets. To address this issue, we propose an effective voting function to smooth the HAS attention across epochs as follows:

$$a_i^{HAS} = \sum_{k=0}^{epoch} \alpha^k \cdot a_{i,k}^{HAS} \quad (4)$$

where $\alpha \in (0, 1)$ denotes the decay rate. $a_{i,k}^{HAS}$ is the HAS attention score for snippet $i$ obtained at the $k$-th training epoch. This function select snippets that consistently exhibit high actionness

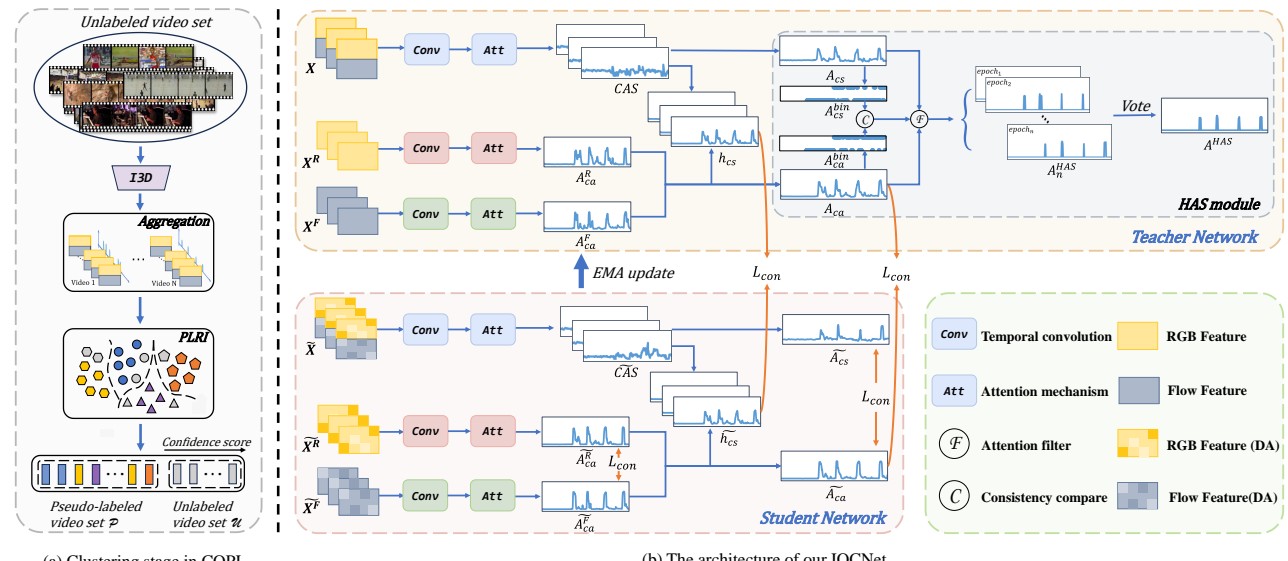

(a) Clustering stage in COPL

(b) The architecture of our IOCNet

Figure 2: Overview of our COPL framework. (a) depicts the clustering stage. The snippet features of each video are aggregated upon HAS attention to create their global feature for spectral clustering. With the PLRI strategy during clustering, the entire set is divided into a pseudolabeled set $P$ and an unlabeled video set $U$. (b) illustrates our proposed IOCNet based on the Teacher-Student architecture. We train IOCNet using both Data-Augmented (DA) features and original features from two video sets. Simultaneously, the HAS module is employed to extract more discriminative snippets for the next itration.

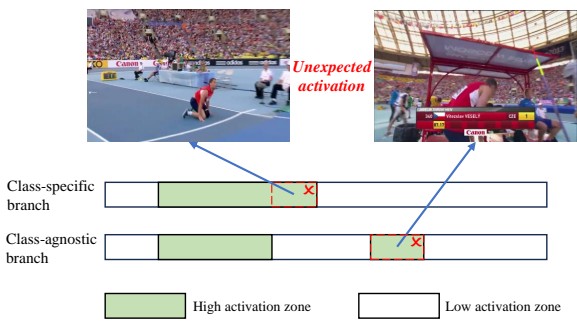

Figure 3: The motivation of HAS. The actionness of two branches exhibits distinct erroneous high activation zones.

throughout the training process. Finally, this smooth HAS attention offers an enhanced and stable perception of actionness, enabling the extraction of more discriminative global video feature as follows:

$$F = L_2 Norm(X^T A^{HAS}) \qquad (5)$$

where $L_2 Norm$ denotes $L_2$ normalization.

## 3.3 Progressive Learning with Representative Instances

Given the global features $F$ of all videos in $V$, we construct a fully-connected graph $\mathcal{G} = \{\mathcal{V}, \mathcal{E}\}$ and perform the spectral clustering [37] to assign $V$ into $C$ clusters as in [10]. The edge weight between video $X_i$ and video $X_j$ is computed as $w_{ij} = \exp(-\frac{dist(F_i, F_j)^2}{2\sigma^2})$, where $\sigma = \frac{1}{N^2} \sum_{i=1}^{N} \sum_{j=1}^{N} dist(F_i, F_j)$. $dist(\cdot, \cdot)$ denotes the Euclidean

distance. After clustering, the pseudolabeled set $P_o$ is constructed. As mentioned above, existing UTAL methods directly adopt the entire set $P_o$ for the later training stage. Due to the inferior clustering in the initial iteration, this operation will inevitably introduce numerous unreliable noisy instances. To address this issue, we introduce the PLRI strategy to obtain a cleaner high-quality pseudolabeled set.

Considering the inherent differences between features of different actions, we craft a metric based on intra-cluster cohesion and inter-cluster separation within the fully connected graph $G$ [34] to filter out less discriminative instances, thereby retaining reliable ones in $P_o$. More Formally, the intra-cluster cohesion is measured as the average distance of video $X_i$ to other videos within their cluster $Y_I$, which is computed as:

$$a_i = \frac{1}{|Y_I|} \sum_{i,j \in C_I, i \neq j} w_{ij} \qquad (6)$$

where $|Y_I|$ represents the cluster size. Similarly, the inter-cluster separation is measured as the average distance between $X_i$ and videos affiliated with its nearest neighboring cluster $Y_J$:

$$b_i = \min_{J \neq I} \frac{1}{|Y_J|} \sum_{j \in Y_J} w_{ij} \qquad (7)$$

The final confidence score $s_i$ is formulated by incorporating the metrics $a_i$ and $b_i$:

$$s_i = \frac{a_i - b_i}{max(a_i, b_i)} \qquad (8)$$

where a higher $s_i$ indicates a stronger intra-cluster cohesion coupled with a larger inter-cluster separation. Essentially, the reliable video with higher confidence of pseudolabel should exhibit a global

representation that is more tightly clustered within their respective action classes, while also demonstrating distinct separation from representations of other actions. Therefore, this metric not only signifies higher confidence in the assigned labels but also reflects the strong discriminative nature of the action-related features.

With this metric, videos in $P_o$ with lower confidence will be removed into an unlabeled set $U$ as follows:

$$U = \{X_i | X_i \in P_o, s_i < \delta\} \quad (9)$$

Furthermore, the refined pseudolabeled set $P$ is constructed as:

$$P = \{(X_i, y_i) | X_i \in P_o, s_i \geq \delta\} \quad (10)$$

where $y_i$ is the action pseudolabel of $X_i$ obtained from clustering.

## 3.4 Inner-Outer Consistency Net

With the refined pseudolabeled video set $P$, the second stage, i.e., training the localization model with $P$, can be conducted. To further facilitate the awareness of high-actionness snippets while taking advantage of unlabeled set $U$, we introduce an IOCNet based on the Teacher-Student architecture [42]. Specifically, since the teacher and student networks have an identical structure, we detail the structure of teacher as an example. For the class-specific branch, $X$, the concatenated feature of $X^R$ and $X^F$, is fed into a convolution layer and a followed attention layer to derive class-specific attention as $CAS = Att(Conv(X))$, where $Att$ comprises several convolution layers and the sigmoid function [22, 53]. Thereafter, $CAS$ is aggregated along the action categories dimension as follows:

$$A_{cs} = Sigmoid(f_{sum}(CAS)) \quad (11)$$

For class-agnostic branch, we separately input $X^R$ and $X^F$ into a temporal convolution layer to obtain class-agnostic attention $A_{ca}^R$ and $A_{ca}^F$ for each respective modality as:

$$A_{ca}^R = Att(Conv(X^R)) \quad (12)$$

Afterwards, the results from both modalities, $A_{ca}^R$ and $A_{ca}^F$, are combined together to generate the class-agnostic attention $A_{ca}$ as:

$$A_{ca} = \frac{A_{ca}^R + A_{ca}^F}{2} \quad (13)$$

After the training of IOCNet, $A_{ca}$ and $A_{cs}$ of each video are fed into HAS module for the generation of the global feature in the next iteration.

### 3.4.1 Training.
The optimization of IOCNet incorporates the following loss:

**a) Weakly-Supervised loss.** With the pseudolabeled set $P$, the common action classification loss can be adopted. Specifically, we first calculate the action selection function [28] as:

$$h_{cs} = \beta \cdot Softmax(CAS) + (1 - \beta)(A_{ca}^R + A_{ca}^F) \quad (14)$$

where $\beta$ balances those two branches. We select top-$l$ action instances with the highest $h_{cs}$ values and generate their video-level predictions $p \in \mathbb{R}^C$ based on the action categories with the maximum values. The action classification loss is computed as:

$$L_{cls} = -\sum_{n=1}^{C} y_n \log(p_n) \quad (15)$$

To optimize the class-agnostic branch, we partition the selected instances into a positive set $S^p$, encompassing instances that match the groundtruth classes and a negative set $S^n$, which includes all other instances. The action selection loss is denoted as:

$$L_{asl} = \frac{1}{|S^p|} \sum_{t \in S^p} \frac{1 - (A_{ca;t}^m)^q}{q} + \frac{1}{|S^n|} \sum_{t \in S^n} \frac{1 - (1 - A_{ca;t}^m)^q}{q} \quad (16)$$

where $m \in \{R, F\}$. The weakly-supervised loss is obtained as:

$$L_{sup} = L_{cls} + L_{asl} \quad (17)$$

**b) Unsupervised loss.** To fully leverage the information in $U$, we introduce unsupervised consistency constraints from both inner- and outer- sources to enhance the discrimination of our network. Specifically, the inner consistency loss include the following two aspects:

**Modality.** The two modalities, RGB and optical flow, are both distinctive and complementary [17]. Within the class-agnostic branch, a late-fusion strategy is employed to adjust the prediction outcomes of both modalities. The consistency constraints can achieve the information supplementation between the two modalities as follows:

$$L_{con}^m = MSE(A_{ca}^R, A_{ca}^F) \quad (18)$$

**Branch.** As discussed in Section 3.2, the actionness generated by the two branches faces distinct zones of erroneous activation. During the training phase, we impose constraints on both branches, compelling them to produce similar actionness sequences as:

$$L_{con}^b = MSE(A_{cs}, A_{ca}) \quad (19)$$

where $MSE$ is the mean square error. The inner consistency constraints are as follows:

$$L_{CI} = \mu_1 L_{con}^m + \mu_2 L_{con}^b \quad (20)$$

Afterwards, we build the outer consistency loss that aims to maintain the output consistency between the teacher and student network. Leveraging temporal feature shift [44], we randomly shift half of $c$ selected channels forward and the other half backward in the student network to create Data-Augmented (DA) features $\tilde{X} = [\tilde{X^R}, \tilde{X^F}]$, contrasting with the teacher network's use of original features. Thereafter, we impose constraints on the outputs of the class-specific and class-agnostic branches of the teacher and student, respectively, which further enhance the temporal action modeling capability of the network as:

$$L_{CO} = MSE(h_{cs}, \tilde{h_{cs}}) + MSE(A_{ca}, \tilde{A_{ca}}) \quad (21)$$

where $h_{cs}$ and $A_{ca}$ are from the student network, $\tilde{h_{cs}}$ and $\tilde{A_{ca}}$ are from the teacher network. Besides, we also employ contrastive loss $L_C$ to enhance snippets features with significant actionness differences between branches [24]. Our unsupervised loss is:

$$L_{un} = L_{CI} + \lambda_1 L_C + \lambda_2 L_{CO} \quad (22)$$

The student network is trained jointly with the sets $P$ and $U$. For the set $P$, all losses are adopted. As for the unlabeled set $U$, only the unsupervised loss is applied. Note that the weights $\phi$ of the teacher model are updated via Exponential Moving Average (EMA) [42] from the corresponding weights of the student $\phi'$. The entire algorithm is provided in Appendix.

*3.4.2 **Inference**.* Following [10, 26], we map the cluster to a specific action category for evaluations. The teacher model is adopted for inference. Firstly, we calculate $h_{cs}$ to obtain action instance positions. Subsequently, we adopt *CAS* to gauge the likelihood of each position corresponding to actions in various categories. After that, we utilize several thresholds to filter out snippets with probabilities greater than $\theta$ to form candidate proposals. Finally, we apply NMS to eliminate redundant proposals. More details are in Appendix.

## 4 EXPERIMENTS

### 4.1 Experiment Settings

*4.1.1 Dataset.* **THUMOS'14** [16]: This dataset comprises 20 action categories with a total of 200 validation videos and 213 test videos. Videos within this dataset encompass diverse action instances, exhibiting a wide range of action durations, spanning from mere seconds to several minutes. Following the common settings [20, 29], we utilize the 200 validation videos for training and the remaining 213 test videos for evaluation.

**ActivityNet v1.2** [1]: This dataset is a large-scale TAL dataset, comprising 9682 videos in total with 100 outdoor action categories. We adopt the same 4819 training videos, 2383 validation videos, and 2480 testing videos as previous methods [20, 29].

*4.1.2 Evaluation Metrics.* The localization performance is evaluated under varying Intersection over Union (IoU) thresholds using the mean Average Precision (mAP) metric, following the standard protocols [1, 16]. As for the evaluation of the clustering results, three widely-used metrics: purity, Normalized Mutual Information score (NMI), and Adjusted Rand Index (ARI) are adopted, on which the better performance is indicated by higher values.

*4.1.3 Implementation Details.* Following the common settings [28, 53], each video stream is first divided into 16-frame non-overlapping snippets, and then the TV-L1 [50] algorithm and the pre-trained I3D network [4] are employed to extract the 1024-dimension RGB and optical flow features. During both training and testing, 750 and 50 snippets for each video in THUMOS'14 and ActivityNet v1.2 are sampled, respectively. For HAS module, we set $\gamma = 20, \alpha = 0.9$ for THUMOS'14 and $\gamma = 2, \alpha = 0.5$ for ActivityNet v1.2. Additionally, we set $\delta = 0$ in Section 3.3 to create refined pseudolabeled and unlabeled video sets. In training, we use $q = 0.7$ for Eq.(16), and instance selection parameters $l = T/8, \beta = 1/3$ for THUMOS'14, and $l = T/2, \beta = 1/2$ for ActivityNet v1.2. We maintain $\mu_1 = 5$, $\mu_2 = 0.1, \lambda_1 = 0.01$ across both datasets, while adjusting $\lambda_2 = 10$ for THUMOS'14 and $\lambda_2 = 1$ for ActivityNet v1.2. Data augmentation employs $c = 512$, with batch sizes of 16 for THUMOS'14 and 128 for ActivityNet v1.2. In addition, we adopt Adam [18] with a learning rate of 1e-4 and a weight decay of 5e-4 to optimize our model over 300 epochs for THUMOS'14 and 30 epochs for ActivityNet v1.2. The total iteration number is set to 3. During testing, the thresholds $\theta \in [0, 1.0 : 0.1]$ are applied to generate proposals. All experiments are run on a Nvidia Tesla A100 GPU.

### 4.2 Comparison With State-of-the-Art Methods

We compare our method with the several state-of-the-art (SOTA) UTAL and WTAL methods across various IoU thresholds in this section. For the weakly supervised setting, COPL solely utilized

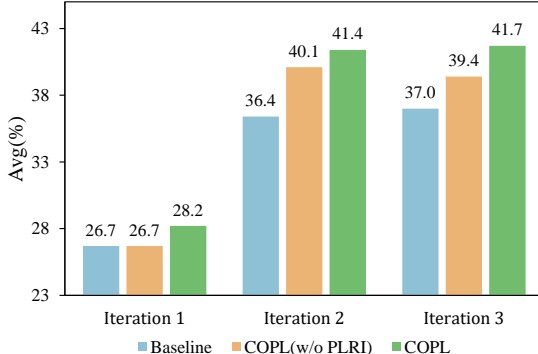

**Figure 4: The results of different variants across iterations.**

the IOCNet, trained once on the entire dataset with the video-level groundtruth labels.

*4.2.1 Results on THUMOS'14 Dataset.* Table 1 illustrates the mAP results of different methods on THUMOS'14 dataset with IoU thresholds ranging from 0.1 to 0.7, with an interval of 0.1. The "Avg" column represents the average mAP results. Our method achieves significant performance in the UTAL scenario, achieving 41.7% on Avg, which marks a notable 1.5% and 6.5% improvement compared with UGCT and APSL. In addition, for mAP@0.5, our COPL method outperforms all other SOTA unsupervised methods by 1.1%, attaining 33.9%.

*4.2.2 Results on ActivityNet v1.2 Dataset.* The localization performance on the ActivityNet v1.2 is presented in Table 2. The "Avg" column represents the average mAP results over the IoU intervals from 0.5 to 0.95, with an interval of 0.05. As depicted in Table 2, even under the UTAL setting, our COPL achieves the best results over all baselines, showing competitive performance compared to several recent weakly supervised methods.

### 4.3 Ablation Studies

In this section, we analyze the roles of different modules in COPL under UTAL conditions. If not specified, the experiments were conducted on THUMOS'14. More results are provided in Appendix.

*4.3.1 Contribution of core modules.* Fig 4 shows the Avg results of three COPL variants w.r.t iteration numbers. The "Baseline" is the COPL (w/o HAS and PLRI) variant that uses the unfiltered class-agnostic attention for clustering and trains with the original pseudolabeled set $P_o$. It is evident that as the iteration proceeds, the localization performance of almost all models shows improvement. Furthermore, the integration of both HAS and PLRI can significantly enhance the performance within the iterations.

*4.3.2 Analysis of HAS module.* This module involves two parts: the action filter of two-branch consistency and the voting function. For the former, we compare the clustering results under the full HAS attention with those under various alternative attention scores, as shown in table 3. For the results of line #2 and #3, the attention filter is conducted on single branch to select high-actionness snippets [26]. Obviously, employing a single branch leads to inferior clustering results. Besides, the clustering advantage of the HAS module

**Table 1: Performance comparison with SOTA methods on THUMOS'14. * denotes the re-implementation for UTAL in [48].**

| Supervision | Method | mAP@IoU (%) | | | | | | | Avg |
|---|---|---|---|---|---|---|---|---|---|
| | | 0.1 | 0.2 | 0.3 | 0.4 | 0.5 | 0.6 | 0.7 | |
| Weakly | TCAM[10], CVPR2020 | - | - | 46.9 | 38.9 | 30.1 | 19.8 | 10.4 | - |
| | CSCL[17], MM2021 | 68.0 | 61.8 | 52.7 | 43.3 | 33.4 | 21.8 | 12.3 | 41.9 |
| | DGCNN[36], MM2022 | 66.3 | 59.9 | 52.3 | 43.2 | 32.8 | 22.1 | 13.1 | 41.3 |
| | TEN[23], MM2022 | 69.7 | 64.5 | 58.1 | 49.9 | 39.6 | 27.3 | 14.2 | 46.1 |
| | UGCT[48], TPAMI2022 | 70.3 | 65.3 | 57.9 | 47.8 | 35.8 | 23.3 | 11.1 | 44.5 |
| | APSL[26], INS2023 | 69.1 | 62.4 | 53.7 | 43.6 | 33.6 | 23.8 | 12.8 | 42.7 |
| | CASE[25], ICCV2023 | 72.3 | - | 59.2 | - | 37.7 | - | 13.7 | 46.2 |
| | AICL[24], AAAI2023 | 73.1 | 67.8 | 58.2 | 48.7 | 36.9 | 25.3 | 14.9 | 46.4 |
| | Wang et al.[46], CVPR2023 | 73.0 | 68.2 | **60.0** | 47.9 | 37.1 | 24.4 | 12.7 | 46.2 |
| | PMIL[33], CVPR2023 | 71.8 | 67.5 | 58.9 | 49.0 | 40.0 | **27.1** | 15.1 | 47.0 |
| | SPCC-Net[35], TMM2024 | 72.6 | 67.3 | 59.4 | 48.7 | 38.3 | 25.6 | 13.4 | 46.5 |
| | ISSF[49], AAAI2024 | 72.4 | 66.9 | 58.4 | 49.7 | **41.8** | 25.5 | 12.8 | 46.8 |
| | COPL | **73.7** | **68.6** | 59.3 | **50.1** | 37.8 | 25.7 | **15.6** | **47.3** |
| Unsupervised | STPN*[31], CVPR2018 | 50.1 | 45.8 | 40.6 | 32.3 | 20.9 | 10.7 | 4.6 | 29.9 |
| | WSAL-BM*[32], CVPR2019 | 57.7 | 52.4 | 46.4 | 37.1 | 26.1 | 16.0 | 6.7 | 34.6 |
| | TSCN*[51], CVPR2020 | 57.1 | 51.6 | 43.9 | 35.3 | 26.0 | 15.7 | 6.0 | 33.7 |
| | TCAM[10], CVPR2020 | - | - | 39.6 | 32.9 | 25.0 | 16.7 | 8.9 | - |
| | UGCT[48], TPAMI2022 | 63.4 | 57.8 | 51.7 | 44.0 | 32.8 | 21.6 | 10.1 | 40.2 |
| | APSL[26], INS2023 | 57.7 | 52.4 | 44.1 | 35.9 | 27.9 | 18.5 | 10.0 | 35.2 |
| | COPL | **65.4** | **60.5** | **52.6** | **44.0** | **33.9** | **22.8** | **12.8** | **41.7** |

**Table 2: Performance comparison with SOTA methods on ActivityNet v1.2. * denotes the re-implementation for UTAL in [48].**

| Supervision | Method | mAP@IoU (%) | | | Avg |
|---|---|---|---|---|---|
| | | 0.5 | 0.75 | 0.95 | |
| Weakly | TCAM[10] | 40.0 | 25.0 | 4.6 | 24.6 |
| | CSCL[17] | 43.8 | 26.9 | 5.6 | 26.9 |
| | DGCNN[36] | 42.0 | 25.8 | 6.0 | 26.2 |
| | TEN[23] | 41.6 | 24.8 | 5.4 | 25.2 |
| | UGCT[48] | 43.1 | 26.6 | 6.1 | 26.9 |
| | APSL[26] | 44.3 | 28.5 | 6.2 | 28.2 |
| | CASE[25] | 43.8 | 27.2 | **6.7** | 27.9 |
| | AICL[24] | 49.6 | 29.1 | 5.9 | 29.9 |
| | PMIL[33] | 44.2 | 26.1 | 5.3 | 26.5 |
| | COPL | **50.2** | **30.2** | 6.5 | **30.7** |
| Unsupervised | STPN*[31] | 28.2 | 16.5 | 3.7 | 16.9 |
| | WSAL-BM*[32] | 28.5 | 17.6 | 4.1 | 17.6 |
| | TSCN*[51] | 22.3 | 13.6 | 2.1 | 13.6 |
| | TCAM[10] | 35.2 | 21.4 | 3.1 | 21.1 |
| | UGCT[48] | 37.4 | 23.8 | 4.9 | 22.7 |
| | APSL[26] | 43.7 | 28.1 | 5.8 | 27.6 |
| | COPL | **48.4** | **28.9** | **6.5** | **29.9** |

over the unfiltered attention on the class-agnostic branch (line #1) also verifies the importance of high actionness for clustering.

**Table 3: Ablation study of HAS module.**

| # | Method | Purity | NMI | ARI |
|---|---|---|---|---|
| 1 | $w/o$ filter | 0.770 | 0.811 | 0.614 |
| 2 | Singe Branch($A_{cs}$) | 0.835 | 0.834 | 0.679 |
| 3 | Singe Branch($A_{ca}$) | 0.850 | 0.849 | 0.693 |
| 4 | HAS | **0.870** | **0.867** | **0.742** |

Moreover, we also analyze the impact of the $\gamma$ in Eq.3, i.e., the selected proportion of high-actionness snippets, on the localization results, as shown in Table 5. A larger $\gamma$ represents less selected snippets. Too small $\gamma$ will cause HAS to introduce too many low-action fragments, which are often associated with background information, consequently leading to a decrease in overall performance. Conversely, when $\gamma$ is large, the total number of selected snippets is too small, which will also hurt the video representation. In our experiment, the best results are obtained when $\gamma$ is set to 20.

For the voting function, as shown in Table 4, we calculate the mean and variance of the respective NMI for COPL and COPL (w/o Vote) variants over their last 10 and 50 epochs. It can be seen that the voting function significantly improves the clustering performance and stability of the model by integrating action information obtained from different epochs.

4.3.3 *Analysis of $\delta$.* In Table 6, we analye the impact of threshold $\delta$ in our PLRI strategy, which balances the quantity and quality of the

**Table 4: Ablation study of the voting function in HAS module.**

| Method | Last epoch | Mean↑ | Variance↓ |
|---|---|---|---|
| $w/o$ Vote | 10 | 0.861 | 0.015 |
|  | 50 | 0.860 | 0.013 |
| $w/$ Vote | 10 | 0.868 | 0.0017 |
|  | 50 | 0.866 | 0.0026 |

**Table 5: Ablation study of the number $\gamma$.**

| $\gamma$ | 5 | 10 | 20 | 30 | 40 |
|---|---|---|---|---|---|
| mAP@Avg | 38.4 | 38.7 | **41.7** | 40.2 | 39.8 |

**Table 6: Analysis of $\delta$ in PLRI module.**

| Method | $\delta$ | THUMOS'14 | | ActivityNet v1.2 | |
|---|---|---|---|---|---|
|  |  | @0.5 | @Avg | @0.75 | @Avg |
| w/o PLRI | - | 32.2 | 40.1 | 28.6 | 28.9 |
| COPL | -0.05 | 32.6 | 41.2 | 28.7 | 29.5 |
|  | 0 | **33.9** | **41.7** | **29.1** | **29.9** |
|  | 0.05 | 33.3 | 41.0 | 28.9 | 29.7 |
|  | 0.10 | 29.9 | 38.2 | 28.3 | 29.0 |

pseudolabeled videos in $P$: the greater the $\delta$, the more strictly the pseudolabeled set $P$ is built. It is evident that when $\delta$ approaches 0, COPL consistently outperforms COPL (w/o PLRI) variant. However, a decrease in localization results is observed when $\delta = 0.10$. This observation highlights that while high-quality labels are beneficial, an insufficient number of instances hinders model learning.

*4.3.4 Analysis of IOCNet.* We incrementally incorporate various components based on supervised loss to assess their impact on IOC-Net's performance, as shown in Table 7. The incorporation of outer consistency constraints $L_{CO}$ enables us to leverage more temporal information, Omitting $L_{CO}$ simplifies the teacher-student network to a single-model structure with all weakly-supervised loss solely on unaugmented data. The setup of using only supervised loss (line #1) yields inferior results, because the low accuracy in class-specific branch activationa will lead to suboptimal clustering performance. Moreover, compared to the variant in line #1, introducing the inner consistency loss $L_{CI}$ enhances the HAS module and joint training on unlabeled video sets, resulting in significant improvements. Further enhancement in discriminative power is achieved through the contrastive learning loss $L_C$. Overall, we can see that all loss functions in IOCNet contributes to the final localization results.

## 4.4 Qualitative analysis.

We present two qualitative results in Fig 5, where the activation scores of our COPL and the Baseline variant are also provided. It is evident that the Baseline variant generates inaccurate predictions in both the action and background segments as in the dashed rectangle. Conversely, our COPL framework demonstrates enhanced feature discrimination, yielding more precise predictions.

**Table 7: Ablation study of loss function.**

| Method | mAP@IoU (%) | | | | Avg |
|---|---|---|---|---|---|
|  | 0.1 | 0.3 | 0.5 | 0.7 |  |
| $L_{cls} + L_{asl}$ | 43.3 | 32.5 | 18.5 | 6.6 | 25.4 |
| $+ L_{CI}$ | 61.1 | 48.2 | 30.0 | 11.4 | 38.1 |
| $+ L_{CI} + L_C$ | 63.7 | 50.9 | 31.7 | 12.2 | 40.1 |
| $+ L_{CI} + L_C + L_{CO}$ | **65.4** | **52.6** | **33.9** | **12.8** | **41.7** |

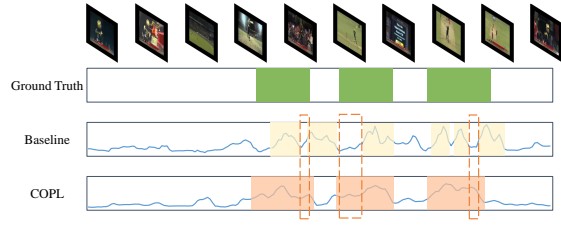

(a) an example video of "CricketShot" action

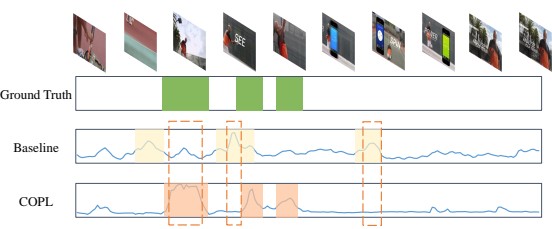

(b) an example video of "TennisSwing" action

**Figure 5: Qualitative comparisons of action localization with Baseline.**

## 5 CONCLUSION

In this paper, we propose a novel consistency-oriented progressive learning framework for UTAL task. Specifically, a high actionness snippet selection module is designed to select most discriminative snippets for better generation of the global video features. To tackle the issue of noisy pseudolabels during training, we propose a PLRI strategy to choose the most reliable pseudolabeled instances based on intra-cluster cohesion and inter-cluster separation. Finally, we propose the teacher-student structured IOCNet that leverages various consistency constraints to improve the temporal action modeling abilties. We also conduct extensive experiments on two public datasets to demonstrate that our COPL framework achieves SOTA UTAL performance and competing weakly-supervised performance. Ablation studies verify the effectiveness of all proposed modules.

**Limitations and Future Work.** While our paper conducts analysis experiments with empirically determined fixed thresholds across various values, our future research will focus on exploring self-adaptive thresholds [45]. Additionally, we are interested in investigating the integration of a deep clustering algorithm [3], moving away from the conventional approach of clustering with original features.

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
