# OpenReview forum: "Revisiting Unsupervised Temporal Action Localization: The Primacy of High-Quality Actionness and Pseudolabels"
_acmmm.org/ACMMM/2024/Conference — MM2024 Poster_

### Official Review · Reviewer_NMrh · 2024-05-23

**Rating:** 6
**Confidence:** 3

**Summary:**

This paper addresses two critical questions in the literature of unsupervised temporal action localization (UTAL):

- What features should the model generate for clustering?
- How to select reliable pseudo-labeled instances?

To answer the first question, this paper introduces a high actionness snippet selection (HAS) module that considers the consistency of class-specific and class-agnostic branches. To address the second question, this paper proposes an IOCNet and a progressive learning paradigm (PLRI), which selects reliable labeled instances for training. Generally, the proposed method can be extended to both weakly supervised temporal action localization (WTAL) and UTAL tasks. Extensive experiments have verified the superiority of the proposed approach, COPL.

**Strengths:**

- This paper exhibits good novelty by addressing two critical issues in the pipeline of UTAL methods.
- The use of case studies to clarify its motivation (e.g., Figure 3) makes the paper easy to understand. Moreover, this paper is well-written and easy to follow.
- To generate high-quality global features and select reliable instances, this paper introduces specific algorithms, which are technically sound.
- The proposed COPL (UTAL) has achieved comparative results compared to advanced WTAL methods. The proposed unsupervised method shows promise for future applications with larger-scale video data.

**Limitations:**

Although this paper has good novelty and is well-written, there are still some drawbacks:

- Mistake in Eq. (6): Since the summation is over both $i$ and $j$, the result $a_i$ should not contain subscript $i$. Otherwise, the summation should not be over $I$. In addition, the $C_I$ in Eq. (6) should be $Y_I$.

- In Eq. (16), this paper incorporates the action selection loss for training, which was proposed by [1] in the WTAL task. It would be better to add this reference.

- The hyper-parameters $\mu_1, \mu_2, \lambda_1, \lambda_2$ in Eqs. (20), (22) should be explained, similar to $\beta$ in Eq. (14).

- Some grammar mistakes: “high-actionness attention that exhibit consistency” in line 266; “a Nvidia” in line 632

[1] Ma, Junwei, Satya Krishna Gorti, Maksims Volkovs, and Guangwei Yu. "Weakly supervised action selection learning in video." In the IEEE Conference on Computer Vision and Pattern Recognition, pp. 7587-7596. 2021.

**Suitability:**

3

---

### Official Review · Reviewer_yPV6 · 2024-05-24

**Rating:** 4
**Confidence:** 3

**Summary:**

This paper tackles unsupervised and weakly supervised temporal action localization. It introduces a module for selecting high-actionness snippets to capture discriminative action-related information effectively. Furthermore, a teacher-student architecture is proposed to enhance discriminative learning. Experimental validation on Thumos and ACNet datasets demonstrates the efficacy of the proposed method.

**Strengths:**

1. The rationale behind selecting discriminative action-related information to improve clustering is sound.
2. The introduction of a teacher-student architecture for enhancing discriminative learning is novel.
3. The effectiveness of the high-actionness snippet selection module is clearly demonstrated through experiments.

**Limitations:**

**Major concerns:**

1. There is a lack of explanation regarding why the proposed method outperforms previous approaches in weakly supervised scenarios. Further analysis is needed to elucidate this superiority.

2. Ablation studies focusing on weakly supervised settings are absent, particularly in explaining why the proposed method exhibits significant improvements in ActivityNet. A detailed analysis of different modules through ablation studies is necessary.

3. Ablation analysis of the teacher-student model is missing. What is the performance in both unsupervised/weakly-supervised settings when either dropping the student network and solely using the teacher model, or dropping the teacher network and solely using the student model?

**Other concerns:**
1. The terms "pseudolabels" and "pseudolabeled" should be replaced with the standard usage "pseudo-labels" and "pseudo-labeled" to align with the existing literature.

**Suitability:**

2

---

### Official Review · Reviewer_Dkji · 2024-05-25

**Rating:** 3
**Confidence:** 3

**Summary:**

In this paper, the authors identify two critical challenges in unsupervised temporal action localization task, and propose a novel yet simple framework named consistency-orientated progressive high actionness learning to generate discriminative global video features for clustering and identify the most reliable and informative instances.

**Strengths:**

(1) The identified two challenges are very critical and deserve research further.
(2) The paper is well-written and good organized.

**Limitations:**

1.Why is the student network not utilized during the inference phase? Does its usage impact performance?
2.What are the differences between the inputs of the teacher network and the student network?
3.Do the authors consider using other clustering methods?
4.The inference efficiency of the proposed method is not compared with the state-of-the-art unsupervised methods.
5.Can you provide more insight into the impact of the HAS module, such as providing visual explanations?
6. In the proposed unsupervised action localization method, there are notable complexities. The reliance on a pre-trained I3D network to distinguish pseudo-labeled or unlabeled instances, along with the introduction of a custom IOCNet, raises concerns about computational overhead and resource utilization. Without empirical data on computational requirements, it is unclear how practical and efficient this approach will be in real-world applications.
7. In terms of the design of IOCNet, the lack of interpretability is noticeable in the placement of the attention structure and certain loss functions, as well as the utilization of some parameters. The complexity in setting hyperparameters also raises questions about the inspiration behind these choices. Were there specific network structures or references that guided the design of IOCNet?
8. The introduction of the dataset is scanned. We only know that these are two action video datasets, using 16 frames. We are not sure if there are any specific downsampling data operations, including the data augmentation mentioned in the article.
9. Despite the adoption of new methods and improvements in effectiveness, the experimental results did not show significant improvement compared to unsupervised and weakly supervised methods in the past two years.

**Suitability:**

2

---

### Meta-Review · Area_Chair_YuCi · 2024-07-02

**Recommendation:** Accept (Poster)
**Confidence:** 5

**Metareview:**

The submission received mixed scores (borderline reject, borderline accept, accept). All reviewers have acknowledge the work addressed two critical challenges in the task of unsupervised action localization. Two reviewers regarded the technical novelty to be clear and sound. Some of the raised concerns (e.g., missing ablation analysis) have been addressed in the rebuttal. The ACs would recommend to accept it and suggest the authors to revise the work according to the suggestions.